# Transporting Tokens: Optimal-Transport View of Parallel LLM Decoding

## Abstract

Autoregressive decoding is a primary bottleneck for large language models (LLMs), as its inherent sequentiality severely limits inference speed. While speculative decoding methods mitigate this via a draft-and-verification pipeline their effectiveness is severely constrained by dependency on draft model quality and availability. We rethink the generation pattern and introduces a novel theoretical perspective by reframing token generation as a predictable state transition process in probability space, formalized through Optimal Transport (OT) theory. We demonstrate that the temporal consistency of hidden states induces a stable transport map, enabling theoretically grounded multi-step prediction. Building on this insight, we develop SHAPE, an OT-based predictor that implements lightweight Sinkhorn iterations. Extensive evaluations across diverse models (e.g., Qwen, Vicuna, LLaMA, DeepSeek) and tasks (text, code, math) show that SHAPE achieves up to 5.23× speedup with minimal quality loss ($\leq 1.2\%$ accuracy drop), empirically validating our distributional transition hypothesis. This work establishes a new theoretical foundation for understanding autoregressive decoding and a practical path toward high-speed generation beyond token-wise limitations.

## 1 Introduction

Large Language Models (LLMs) have become the cornerstone of modern artificial intelligence, achieving remarkable success across tasks ranging from natural language understanding to text generation Mo et al. (2024); Wu (2024); Li et al. (2024a); Shu et al. (2024); Thakur et al. (2024). LLMs of varying scales have been widely deployed in cloud server clusters (e.g., GPT-4 OpenAI et al. (2024), Llama3 Grattafiori et al. (2024), and Grok1 xAI (2024)) and edge devices (e.g., the 6B-parameter GPT-3 and 7B-parameter LLaMA-2 variants as lightweight LLMs Lu et al. (2024); Sun et al. (2020)). With their increasing adoption in search Wang et al. (2024) and conversational AI Ouyang et al. (2022), there is a growing demand for low-latency long-sequence generation, making the optimization of effective token generation rate under constrained computational resources a critical research challenge.

Unfortunately, both cloud-based large models and edge-side small models rely on autoregressive token-by-token generation, which requires sequential computation of each token without parallelization. Additionally, the quadratic complexity of attention mechanisms with respect to context length exacerbates the issue. The standard autoregressive decoding used in existing LLMs suffers from inherent inefficiencies Touvron et al. (2023); Jiang et al. (2023)—generation time scales linearly with both context length and model size, and its sequential nature leads to cumulative latency. Our benchmarking experiments across diverse models reveal that larger model sizes and longer context lengths lead to significantly higher per-token latency. This cost is compounded by the sequential nature of decoding, highlighting the urgent need for optimization to achieve practical deployment efficiency. Comprehensive results are presented in Appendix B.

Speculative decoding Cai et al. (2024) addresses this by introducing a fast draft model to predict multiple tokens in advance, followed by verification from the target model. However this two-stage draft-and-verification paradigm still incurs sequential latency and is highly sensitive to the quality of draft models. Lookahead Fu (2023) and Medusa Cai et al. (2024) reduce decoding time using n-gram

heuristics or shallow predictors, but their limited accuracy (e.g., 0.6 for Medusa) results in suboptimal speedup. EAGLE Li et al. (2024b) improves draft accuracy by leveraging hidden-state features, achieving better acceleration, yet it remains draft-model-dependent, introducing overhead and limiting scalability across diverse model configurations. CLLMs Kou et al. (2024) accelerate decoding by directly predicting future token distributions via conditional probabilities, enabling parallel generation. However, they require fine-tuning parts of the original model, increasing training costs, and while particularly effective for mathematical reasoning, they exhibit limited stability in long-form generation. In contrast, our approach reconceptualizes decoding itself through a distributional lens. In this work, we propose a paradigm shift by reconceptualizing token generation as a *distributional transition process*. Our key insight stems from the empirical observation that hidden states exhibit strong temporal consistency during decoding—consecutive states maintain high semantic similarity with a predictable lower bound. This regularity suggests that token generation follows a structured evolution in probability space, a perspective we formalize through optimal transport (OT) theory.

By modeling the transition between successive token distributions as a mass transport problem, where the semantic similarity between hidden states induces a stable OT map. To empirically validate this theoretical framework, we develop **SHAPE** (Step-ahead Hidden-state Accelerated Prediction Engine) as a concrete instantiation of our OT-based perspective. SHAPE operationalizes the theoretical transport maps by learning lightweight operators between hidden states, enabling parallel token prediction without auxiliary draft models. The empirical success of SHAPE—achieving substantial speedups while maintaining quality—serves as strong evidence for the correctness of our underlying theoretical insight: that token generation can indeed be understood as a predictable transport process in probability space. We evaluated SHAPE on a range of models—including Qwen, Vicuna, LLaMA, and DeepSeek—across general language (WikiText, Alpaca, MT-Bench) and reasoning-heavy tasks (MATH500, AIME24, LiveCodeBench v5). The results show that SHAPE achieves speedups of up to 5.23× while maintaining output quality within a minimal margin of degradation ($\leq 1.2\%$ accuracy drop on reasoning tasks). In comparative experiments, SHAPE consistently outperforms existing acceleration methods: it surpasses EAGLE3 by 1.1x, Medusa-1 by 2.1×, and Medusa-2 by 1.6× across different models and datasets.

Beyond performance, this work makes the following key contributions:

- **A Novel Theoretical Foundation**: We introduce a paradigm shift by reconceptualizing token generation as a predictable transition of probability distributions. This perspective is rigorously formalized through Optimal Transport theory and validated empirically, establishing a new principled understanding of decoding dynamics.
- **A Practical, Plug-and-Play Predictor**: We develop SHAPE, a lightweight prediction engine that operationalizes this theory. Crucially, SHAPE requires no modifications to the base LLM's parameters, offering a draft-free, plug-and-play solution for immediate deployment that significantly enhances decoding efficiency.
- **Scalability to Arbitrary Future Steps.** SHAPE generalizes to predict hidden states at arbitrary future time steps (e.g., $t+1$, $t+2$, $t+3$), providing greater flexibility for long-sequence generation tasks. This scalability supports diverse applications with varying sequence lengths and complexity.

By fundamentally rethinking the decoding process rather than optimizing within its constraints, this work opens new directions for efficient LLM inference.

## 2 FROM STATE SIMILARITY TO DISTRIBUTIONAL TRANSITION

### 2.1 SEMANTIC SIMILARITY OF HIDDEN STATES

Building on recent work that recognizes the regularity of hidden-state sequences Li et al. (2024b) and their utility for parallel prediction Cai et al. (2024), we systematically analyze the temporal correlations between consecutive hidden states during autoregressive decoding. Let $\mathbf{h}_t \in \mathbb{R}^H$ denote the final-layer hidden state at decoding step $t$. We quantify the *semantic consistency* between states at steps $t$ and $t+n$ using cosine similarity:

$$\text{sc}(\mathbf{h}_t, \mathbf{h}_{t+n}) = \frac{\mathbf{h}_t \cdot \mathbf{h}_{t+n}}{\|\mathbf{h}_t\|_2 \cdot \|\mathbf{h}_{t+n}\|_2} \tag{1}$$

Table 1: Hidden State Cosine Similarity Across Models and Domains

| Model | ShareGPT (Text) | | | The Stack (Code) | | |
|---|---|---|---|---|---|---|
| | n=1 | n=2 | n=3 | n=1 | n=2 | n=3 |
| Qwen-7B | 0.83 | 0.78 | 0.75 | 0.63 | 0.53 | 0.50 |
| Qwen-32B | 0.85 | 0.81 | 0.78 | 0.75 | 0.65 | 0.65 |
| Qwen-72B | 0.91 | 0.86 | 0.82 | 0.83 | 0.73 | 0.71 |
| Llama-7B | 0.64 | 0.57 | 0.54 | 0.52 | 0.46 | 0.44 |
| Llama-33B | 0.90 | 0.84 | 0.78 | 0.80 | 0.74 | 0.68 |
| Llama-70B | 0.91 | 0.86 | 0.81 | 0.82 | 0.76 | 0.70 |

**Experimental Settings**   We conduct experiments on two representative domains: conversational text using the ShareGPT dataset and code generation using The Stack dataset. For each domain, we use a context length of 2048 tokens and generate sequences of 512 tokens. All experiments are performed on NVIDIA A100 80GB GPUs, with models ranging from 7B to 72B parameters. For each model-dataset combination, we compute semantic consistency between hidden states at positions $t$ and $t + n$ across 1000 randomly sampled sequences, reporting the 95th percentile values across all valid token positions to ensure statistical significance and capture the lower bound of similarity distribution.

As demonstrated in Table 1, during token generation, hidden states feature exhibit pronounced temporal smoothness and strong semantic consistency: our quantitative analysis **across all valid token positions** shows that for adjacent steps ($n=1, 2, 3$) at least 95% of positions satisfy $\mathrm{SC}(\mathbf{h}_t, \mathbf{h}_{t+n}) \geq \tau$ with $\tau=0.5$, indicating high-probability local stability of the representation. We also observe a consistent pattern of text > code and larger > smaller models, reinforcing that token transitions are smooth and predictable rather than erratic.

The observed consistency persists even in challenging scenarios with potential semantic transitions between dialogue turns in ShareGPT and code blocks in The Stack—demonstrating the generalization of this property across domains with different structural characteristics. This empirical finding suggests that transitions between consecutive token distributions are both small and structured: the changes are concentrated along semantically meaningful directions rather than arbitrary noise, constrained by linguistic coherence in text and syntactic regularities in code. As a result, the generative process of large language models behaves like a smooth dynamical system in a latent state space, where each new token constitutes a predictable, low-dimensional adjustment to the current semantic state rather than a radical reconstruction—providing a stable foundation for predictive modeling and multi-step forecasting.

## 2.2 MODELING TOKEN GENERATION WITH OPTIMAL TRANSPORT

Building upon the empirical observation of strong temporal consistency in hidden states, we introduce a theoretical perspective that elevates token generation to a *structured probability flow* problem. This formulation recognizes that the evolution of hidden states imposes geometric constraints on how token distributions change over time.

Such constraints are not captured by conventional next-token prediction objectives. In contrast, optimal transport (OT) offers a mathematically grounded framework for modeling distributional evolution under minimal geometric distortion. By embedding token generation within the OT formalism, we uncover a deeper structure underlying autoregressive decoding: distribution transitions follow low-cost paths governed by hidden-state continuity. This perspective is not merely as a re-description, but as a foundation for building new path of token evolution, deriving stability guarantees, and enabling multi-step step ahead generation. In short, OT transforms our understanding of decoding from a static pointwise prediction problem to a dynamic, diffusion like, geometry-aware process.

To mathematically capture this structured diffusion evolution, we represent the token distribution as a discrete measure in the hidden state features space:

$$\mu_t = \sum_{i=1}^{V} \mathbf{p}_t(i)\, \delta_{\mathbf{E}_i}, \quad \text{where } \mathbf{p}_t = \mathrm{softmax}(W\mathbf{h}_t/\tau_s).$$

The transition from $\mu_t$ to $\mu_{t+n}$ is then formulated as an *entropic-regularized optimal transport* problem:

$$\Pi_t^\star = \arg \min_{\Pi \mathbf{1} = \mathbf{p}_t, \, \Pi^\top \mathbf{1} = \mathbf{p}_{t+n}} \langle \Pi, C \rangle + \varepsilon \, \mathrm{KL}(\Pi \| \mathbf{p}_t \mathbf{p}_{t+n}^\top),$$

where the cost matrix $C \in \mathbb{R}^{V \times V}$ is defined by the squared Euclidean distance between token embeddings ($C_{ij} = \|\mathbf{E}_i - \mathbf{E}_j\|^2$), and $\varepsilon > 0$ is the regularization strength.

Crucially, the observed semantic consistency provides theoretical guarantees for this formulation. As proven in Lemma C.2 (Appendix), the Wasserstein distance between consecutive distributions is bounded by:

$$W_c(\mu_t, \mu_{t+n}) \leq \bar{L}\sqrt{1 - \mathrm{SC}(\mathbf{h}_t, \mathbf{h}_{t+n})} \leq \bar{L}\sqrt{1 - \tau},$$

which ensures the existence and uniqueness of the optimal coupling $\Pi_t^\star$ (Proposition C.3, Appendix).

The row-normalized optimal coupling $K_t = \mathrm{diag}(\Pi_t^\star \mathbf{1})^{-1}\Pi_t^\star$ defines a principled stochastic transition matrix that characterizes the distributional evolution:

$$\mathbf{p}_{t+n} = K_t^\top \mathbf{p}_t.$$

This formulation casts token generation as a path-following process in the probability simplex, where the temporal stability of hidden states ensures the stability of the transport map. This theoretical insight forms the cornerstone of the SHAPE method, providing a mathematically sound framework for analyzing and intervening in the generation process. Complete proofs and detailed analysis are provided in Appendix C.

# 3 SHAPE: AN OT-GUIDED MULTI-TOKEN PREDICTOR

To validate and operationalize the OT-based transition view, we propose **SHAPE** (Step-ahead Hidden-state Accelerated Prediction Engine), a **draft-free**, plug-and-play framework for parallel decoding. As shown in Figure 1. SHAPE consists of two key components: **Step-ahead Hidden State Prediction** and **Tree Rejection Sampling**. The core design of the framework focuses on capturing the semantic correlation of hidden states by capturing temporal features and training a predictor to approximate future hidden states. With tree reject sampling select the longest accepted prefix in parallel dynamically, so we can get $\alpha$ accepted token in one LLM forward to achieve parallel acceleration.

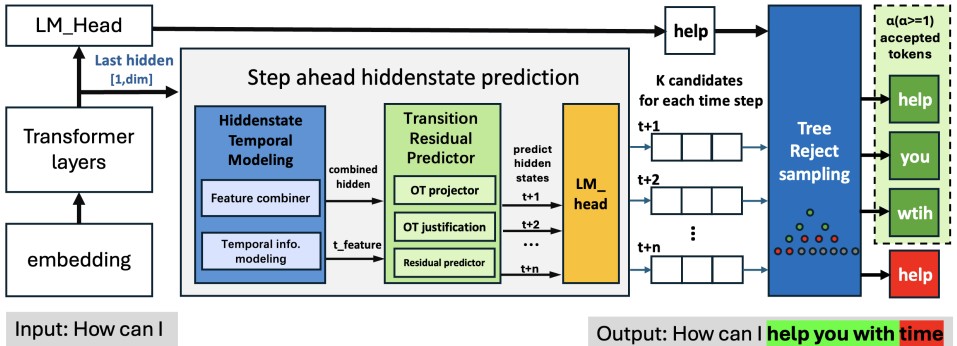

Figure 1: Illustration of the SHAPE (Step-ahead Hidden-state Accelerated Prediction Engine) framework. SHAPE leverages strong temporal correlations in hidden states to predict multiple future tokens by modeling hidden state transitions. It includes temporal modeling and residual predictors for hidden state prediction, followed by edge-to-edge LM head training to generate multiple candidates for each future step. SHAPE uses tree-based rejection sampling to select optimal token candidates at each time step, enabling efficient multi-token generation without a draft model.

## 3.1 HIDDEN STATE SEMANTIC CORRELATION MODELING

The main structure of step-ahead hidden state prediction is shown in Figure 2, with three main trainable components. To first extract features in hidden state temporal modeling, the hidden states

from the current and previous three-time steps are concatenated and passed through a series of transformations, including linear projections, layer normalization, activation functions, and dropout. These steps capture temporal dependencies and refine the features, resulting in a final representation that effectively encodes the relationships between the time steps.

## 3.2 STEP AHEAD HIDDEN STATE RESIDUAL PREDICTOR

### 3.2.1 PREDICTOR CONSTRUCTION

Predicting hidden states directly in the full transformer dimension is challenging. To improve stability, we adopt a residual-based transition model: instead of predicting the entire hidden state, the predictor learns the delta between consecutive states. An adaptive gating mechanism (a Linear–Sigmoid network) dynamically scales the predicted residual based on both the current hidden state and the predicted change, effectively controlling uncertainty and preventing error accumulation in multi-step prediction. Beyond residual modeling, we optionally introduce an Optimal Transport (OT) refinement module to further regularize the transition between $H_t$ and $H_{t+n}$. When enabled, the refinement consists of three lightweight stages:

**(1) Dimensionality Reduction.** A learned projection $P_1 : \mathbb{R}^H \to \mathbb{R}^d$ compresses hidden states into a lower-dimensional space:

$$h_t^d = P_1(H_t), \qquad h_{t+n}^d = P_1(H_{t+n}).$$

**(2) OT-based Alignment.** The reduced states are normalized into distributions:

$$p = \mathrm{softmax}(h_t^d), \qquad q = \mathrm{softmax}(h_{t+n}^d),$$

and aligned via entropy-regularized optimal transport:

$$\min_T \langle T, C \rangle + \varepsilon H(T) \quad \text{s.t.} \quad T\mathbf{1} = p, \quad T^\top \mathbf{1} = q.$$

The cost matrix $C$ reflects semantic discrepancy in the reduced space, and the entropy term prevents overly sparse or unstable transport plans. Since $d \ll H$, the transport computation is efficient.

**(3) Dimension Recovery.** The aligned representation is then projected back to the original space via $P_2 : \mathbb{R}^d \to \mathbb{R}^H$:

$$H_{t+n}^{\mathrm{OT}} = P_2(T^\top \mathbf{1}).$$

Finally, the refined hidden state combines the raw residual predictor output and the OT-aligned result:

$$H_{t+n} = (1 - \alpha)H_{t+n}^{\mathrm{raw}} + \alpha H_{t+n}^{\mathrm{OT}},$$

where $\alpha \in [0, 1]$ controls the strength of OT refinement.

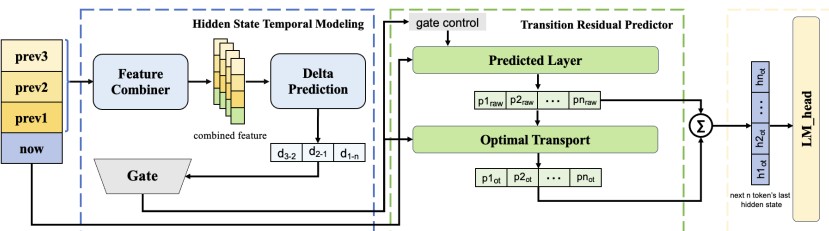

Figure 2: Step-ahead hidden state predictor: temporal modeling (blue), residual prediction (green), and LM-head projection (yellow).

### 3.2.2 PREDICTOR TRAINING

The hidden state predictor architecture is designed to maintain dimensional consistency with the source large language model, preserving the original hidden state dimensionality. The training procedure utilizes optimal transport learning ($\alpha = 0.5$, $\epsilon = 0.1$) to enhance multi-step prediction accuracy. The training corpus consists of preprocessed hidden states extracted from both English ShareGPT conversational data and Chinese THUC_News articles, enabling bilingual prediction capabilities. The training was conducted using AdamW optimization with mixed-precision computation, incorporating uniform noise augmentation (std = 0.2) to improve model robustness. Input sequences were truncated at 2048 tokens to maintain computational efficiency with batch size = 16. The training objective combined two loss terms:

**Hidden State Loss** This loss optimizes the consistency between predicted hidden states $\hat{h}t+n$ and target hidden states $ht+n$ using mean squared error:

$$\mathcal{L}_{\text{hidden}} = \frac{1}{N} \sum_{i=1}^{N} \left\| \hat{h}_{t+n}^i - h_{t+n}^i \right\|_2^2 \tag{2}$$

where $N$ is the sample size.

**Token Distribution Loss** This cross-entropy loss ensures alignment between predicted and target token distributions:

$$\mathcal{L}_{\text{token}} = -\frac{1}{N} \sum_{i=1}^{N} \sum_{j=1}^{V} p_{\text{target}}^i(j) \log p_{\text{output}}^i(j) \tag{3}$$

where $p_{\text{target}}(j)$ and $p_{\text{output}}(j)$ represent the target and predicted token distributions respectively, and $V$ is the vocabulary size.

### 3.3 TREE REJECT SAMPLING

**Tree Rejection Sampling** generates multiple candidate paths for the next $N$ tokens at time step $t$, forming a tree structure of width $k$ and depth $N$ (thus producing $k^N$ candidate paths). The model then computes the joint probabilities of these paths in parallel. Low-probability paths are rejected based on a predefined acceptance threshold, and the remaining paths are merged by selecting the longest valid prefix. This design balances generation diversity and quality by exploring multiple future branches in a single forward pass. Detailed algorithm implementation is shown in Appendix.

## 4 EXPERIMENTS

We evaluate SHAPE across major LLM families—including Vicuna(7B/13B), LLaMA2-Chat(7B/13B/70B), Qwen (7B/14B/72B), and recent long-chain reasoning models such as Qwen3 and DeepSeek-R1 both efficiency and output quality. Our benchmarks span three categories: (1) general text generation (Alpaca, WikiLingua; evaluated with PPL), (2) knowledge and reasoning tasks (MMLU accuracy and MT-Bench scores), and (3) challenging long-context reasoning datasets (MATH500, AIME24, LiveCodeBench v5). All experiments are conducted on NVIDIA A100 80GB GPUs under consistent settings to ensure fair comparison.

### 4.1 EFFICIENCY

We present a comprehensive comparison of SHAPE against Lookahead, Medusa-1, Medusa-2, and EAGLE3 under both temperature 0 and 1 across eight major model families. As shown in Table 2, SHAPE consistently achieves the highest or near-highest speedups across all datasets and temperatures.

Compared with Lookahead, SHAPE delivers substantially larger gains, typically improving speed by **1.5×–2×**. Relative to Medusa-1 and Medusa-2, SHAPE provides clear improvements under every model, with speedups exceeding both methods in all Alpaca, Wiki, and MT-Bench settings. SHAPE also closely tracks or surpasses EAGLE3 across all model scales: on Qwen-7B/14B/72B,

Table 2: Speedup comparison among Lookahead, Medusa-1, Medusa-2, EAGLE3, and SHAPE under Temperature = 0 and 1 across datasets.

| Model | Method | Alpaca$_0$ | Alpaca$_1$ | Wiki$_0$ | Wiki$_1$ | MT$_0$ | MT$_1$ | Mean$_0$ | Mean$_1$ |
|---|---|---|---|---|---|---|---|---|---|
| Qwen-7B | Lookahead | 2.71× | 2.43× | 2.50× | 2.27× | 3.05× | 2.66× | 2.75× | 2.45× |
| | Medusa-1 | 1.77× | 1.59× | 1.87× | 1.68× | 2.02× | 1.82× | 1.89× | 1.70× |
| | Medusa-2 | 2.04× | 1.80× | 2.16× | 1.90× | 2.49× | 2.19× | 2.23× | 1.96× |
| | EAGLE3 | **4.13×** | 3.72× | 4.05× | 3.65× | 4.34× | 3.73× | 4.17× | 3.70× |
| | SHAPE | 4.12× | **3.83×** | **4.10×** | **3.81×** | **4.53×** | **4.03×** | **4.25×** | **3.89×** |
| Qwen-14B | Lookahead | 2.46× | 2.18× | 2.38× | 2.11× | 3.21× | 2.78× | 2.68× | 2.36× |
| | Medusa-1 | 2.01× | 1.81× | 2.03× | 1.83× | 2.11× | 1.90× | 2.05× | 1.85× |
| | Medusa-2 | 2.29× | 2.02× | 2.28× | 2.01× | 2.51× | 2.21× | 2.36× | 2.08× |
| | EAGLE3 | **4.01×** | 3.53× | 3.97× | 3.57× | 5.11× | 4.34× | 4.36× | 3.81× |
| | SHAPE | 3.90× | **3.55×** | **4.03×** | **3.71×** | **5.23×** | **4.60×** | **4.39×** | **3.95×** |
| Qwen-72B | Lookahead | 3.10× | 2.82× | 3.05× | 2.77× | 3.65× | 3.21× | 3.27× | 2.93× |
| | Medusa-1 | 2.18× | 1.96× | 2.12× | 1.91× | 2.52× | 2.27× | 2.27× | 2.05× |
| | Medusa-2 | 3.15× | 2.77× | 3.07× | 2.70× | 3.64× | 3.20× | 3.29× | 2.89× |
| | EAGLE3 | 4.85× | 4.35× | 4.72× | 4.21× | 5.60× | 4.95× | 5.06× | 4.50× |
| | SHAPE | **4.92×** | **4.48×** | **4.80×** | **4.33×** | **5.73×** | **5.18×** | **5.15×** | **4.66×** |
| Llama-7B | Lookahead | 2.89× | 2.55× | 2.76× | 2.43× | 3.30× | 2.88× | 2.98× | 2.62× |
| | Medusa-1 | 1.88× | 1.69× | 1.85× | 1.67× | 2.09× | 1.88× | 1.94× | 1.75× |
| | Medusa-2 | 3.01× | 2.65× | 3.05× | 2.68× | 2.58× | 2.27× | 2.88× | 2.53× |
| | EAGLE3 | 4.20× | 3.78× | 4.01× | 3.61× | 4.65× | 4.00× | 4.29× | 3.80× |
| | SHAPE | **4.23×** | **3.93×** | **4.11×** | **3.82×** | **4.73×** | **4.21×** | **4.36×** | **3.99×** |
| Llama-13B | Lookahead | 2.63× | 2.38× | 2.58× | 2.34× | 3.41× | 2.93× | 2.87× | 2.55× |
| | Medusa-1 | 2.03× | 1.83× | 2.01× | 1.81× | 2.13× | 1.92× | 2.06× | 1.85× |
| | Medusa-2 | 3.15× | 2.77× | 3.12× | 2.75× | 2.76× | 2.43× | 3.01× | 2.65× |
| | EAGLE3 | 4.12× | 3.71× | 4.12× | 3.71× | 4.78× | 4.11× | 4.34× | 3.84× |
| | SHAPE | **4.13×** | **3.80×** | **4.15×** | **3.86×** | **5.01×** | **4.51×** | **4.43×** | **4.06×** |
| Llama-70B | Lookahead | 3.25× | 2.95× | 3.21× | 2.92× | 3.78× | 3.36× | 3.41× | 3.08× |
| | Medusa-1 | 2.26× | 2.03× | 2.21× | 1.99× | 2.62× | 2.36× | 2.36× | 2.13× |
| | Medusa-2 | 3.26× | 2.87× | 3.19× | 2.81× | 3.78× | 3.33× | 3.41× | 3.00× |
| | EAGLE3 | 5.02× | 4.55× | 4.91× | 4.48× | 5.82× | 5.14× | 5.25× | 4.72× |
| | SHAPE | **5.10×** | **4.70×** | **5.00×** | **4.62×** | **5.95×** | **5.38×** | **5.35×** | **4.90×** |
| Vicuna-7B | Lookahead | 2.52× | 2.29× | 2.48× | 2.25× | 3.19× | 2.80× | 2.73× | 2.45× |
| | Medusa-1 | 1.79× | 1.61× | 1.84× | 1.66× | 2.18× | 1.96× | 1.94× | 1.74× |
| | Medusa-2 | 2.88× | 2.53× | 2.91× | 2.56× | 2.83× | 2.49× | 2.87× | 2.53× |
| | EAGLE3 | 3.95× | 3.52× | 3.90× | 3.51× | 5.11× | 4.34× | 4.32× | 3.79× |
| | SHAPE | **3.98×** | **3.66×** | **3.95×** | **3.67×** | **5.13×** | **4.57×** | **4.35×** | **3.97×** |
| Vicuna-13B | Lookahead | 2.60× | 2.33× | 2.56× | 2.30× | 3.26× | 2.89× | 2.81× | 2.51× |
| | Medusa-1 | 2.05× | 1.84× | 2.07× | 1.86× | 2.33× | 2.10× | 2.15× | 1.93× |
| | Medusa-2 | 2.86× | 2.52× | 2.89× | 2.54× | 2.85× | 2.51× | 2.87× | 2.52× |
| | EAGLE3 | **4.05×** | 3.65× | 4.00× | 3.60× | 4.57× | 3.93× | 4.21× | 3.73× |
| | SHAPE | 4.00× | **3.72×** | **4.07×** | **3.79×** | **5.13×** | **4.62×** | **4.40×** | **4.04×** |

Llama-7B/13B/70B, and Vicuna-7B/13B, SHAPE achieves the best overall mean speedup in both temperature settings. Notably, SHAPE consistently improves over EAGLE3 at temperature 1, where speculative methods generally become less stable. Overall, the results demonstrate that SHAPE provides the most stable and highest average speedup across all model families, datasets, and sampling temperatures, outperforming prior speculative decoding and multi-head prediction baselines in a uniform evaluation setup.

We evaluate SHAPE's inference efficiency under varying batch sizes to assess its practicality in real-world deployment scenarios. Using the MT-Bench dataset on the Qwen2-7B model, we compare SHAPE against EAGLE-3, with vLLM without speculative sampling as the baseline. As shown in Table 3, SHAPE consistently outperforms EAGLE-3 across all batch sizes, demonstrating superior scalability and efficiency in practical batch processing environments. The results confirm that

while both methods exhibit reduced relative gains at larger batch sizes due to increased baseline parallelism, SHAPE maintains a consistent performance advantage.

Table 3: Speedup ratios at different batch sizes

| Method | BS = 2 | BS = 4 | BS = 8 | BS = 16 | BS = 24 |
|---|---|---|---|---|---|
| EAGLE-3 | 1.73× | 1.65× | 1.52× | 1.43× | 1.39× |
| SHAPE | 1.92× | 1.75× | 1.61× | 1.52× | 1.41× |

## 4.2 QUALITY EVALUATION

We evaluate generation quality from three perspectives: (1) general performance on standard benchmarks, (2) token-level prediction accuracy and semantic consistency, and (3) performance on reasoning-intensive and long-context tasks.

**General Performance Evaluation.** We evaluate generation quality using PPL on Alpaca and Wiki-Text, MMLU accuracy for reasoning, and MT-Bench for conversational ability. As shown in Table 4, SHAPE maintains output quality across all model families, with PPL remaining close to baseline and MMLU/MT-Bench varying within 1–2%. These results indicate that SHAPE preserves model utility while providing significant decoding acceleration.

Table 4: Performance comparison between Vanilla and SHAPE-accelerated models across benchmarks.

| Model | Alpaca (PPL) | WikiText (PPL) | MMLU-5shot (Acc) | MT-Bench (Score) |
|---|---|---|---|---|
| Qwen-7B | 11.49 / 11.9 | 11.89 / 12.1 | 70.5 / 68.79 | 8.41 / 8.56 |
| Qwen-14B | 12.30 / 11.8 | 11.92 / 11.7 | 66.3 / 64.78 | 9.08 / 8.85 |
| Qwen-72B | 10.95 / 11.2 | 10.88 / 11.0 | 75.6 / 74.9 | 9.62 / 9.55 |
| Llama-7B | 11.76 / 12.2 | 12.77 / 12.4 | 46.2 / 44.32 | 6.27 / 6.43 |
| Llama-13B | 12.67 / 12.3 | 11.94 / 12.5 | 55.0 / 56.38 | 7.05 / 6.89 |
| Llama-70B | 10.88 / 11.1 | 10.72 / 11.0 | 67.5 / 66.8 | 8.92 / 8.85 |
| Vicuna-7B | 11.58 / 12.1 | 12.83 / 12.4 | 48.2 / 48.55 | 6.69 / 6.88 |
| Vicuna-13B | 12.06 / 11.7 | 11.63 / 12.0 | 55.28 / 58.42 | 6.81 / 6.97 |

**Reasoning-Intensive Task Evaluation.** As shown in Table 5, SHAPE maintains near-identical accuracy compared to vanilla decoding across all models and tasks. On MATH500, accuracy differences are within 0.3%, while AIME24 and LiveCodeBench v5 show maximum deviations of 1.3% and 0.8% respectively. These results confirm SHAPE's robustness on reasoning-heavy, long-chain tasks while delivering 4-5× speedups.

Table 5: Accuracy comparison on reasoning-intensive tasks (Vanilla / SHAPE)

| Model | MATH500 | AIME24 | LiveCodeBench v5 |
|---|---|---|---|
| Qwen3-32B | 97.2 / 97.16 | 81.4 / 80.8 | 65.7 / 65.3 |
| Qwen3-8B | 97.4 / 97.1 | 76.0 / 74.7 | 57.5 / 56.9 |
| DeepSeek-R1-Distill-Qwen-32B | 94.3 / 93.2 | 72.6 / 72.2 | 54.5 / 54.1 |
| DeepSeek-R1-Distill-Qwen-14B | 93.9 / 92.1 | 69.7 / 68.9 | 45.5 / 44.7 |

**Token-Level Analysis.** Supplementary evaluations (Appendix Tables 12 and 10) show SHAPE achieves token prediction accuracy of 0.85-0.92 for 1-3 token lookahead, with lower perplexity compared to alternative acceleration methods. Semantic similarity metrics (BERTScore and embedding cosine distance) confirm strong alignment with standard decoding outputs.

## 4.3 ABLATION STUDY

### 4.3.1 EFFECTIVENESS OF OPTIMAL TRANSPORT

SHAPE employs optimal transport (OT) to model hidden state transitions, motivated by our observation that transformer hidden states maintain a minimum level of similarity between tokens at $t$ and $t+n$. This "baseline similarity" indicates a theoretically valid pathway for transferring hidden states through optimal transport. Unlike autoregressive models that predict step-by-step, our OT approach captures global transition patterns by finding the optimal mapping to future states ($t+n$). To validate the effectiveness of OT over simpler alternatives, we conducted comparative experiments replacing the OT mapping with an affine transformation of the same dimensionality ($d = 128$). Table 6 presents the results on Llama-7B using the Alpaca dataset with TRS configuration ($N = 3, K = 3$).

Table 6: Comparison of OT with affine transformation and analysis of different dimensionalities on Llama-7B (Alpaca). Baseline AR decoding achieves PPL=11.9.

| Method | PPL | Speedup |
|---|---|---|
| Affine ($d$=128) | 18.4 | 3.87× |
| **OT ($d$=128)** | **12.2** | 3.21× |

| $d$ **Value** | **PPL** | **Speedup** |
|---|---|---|
| 32 | 17.3 | 3.87× |
| 64 | 16.1 | 3.66× |
| **128** | **12.2** | 3.21× |
| 4096 (full) | 12.1 | 2.67× |

The results demonstrate that OT significantly outperforms simple affine transformations, reducing perplexity from 18.4 to 12.2 - approaching the baseline AR performance of 11.9. This validates our hypothesis that OT's ability to find optimal global mappings is crucial for accurate multi-step prediction. Furthermore, we analyzed the impact of dimensionality $d$ on OT performance. As shown in Table 6 (right), increasing $d$ from 32 to 128 consistently improves perplexity, with the most significant gains occurring at $d = 128$. Interestingly, using the full dimensionality ($d = 4096$) provides minimal perplexity improvement (12.1 vs 12.2) while reducing speedup by 17%, confirming that our low-dimensional OT approach effectively captures essential transition patterns.

### 4.3.2 EFFECTIVENESS OF TREE REJECTION SAMPLING

We provide a unified analysis of SHAPE decoding behavior and the proposed Tree Rejection Sampling (TRS) mechanism in Table 7. The top block reports the one-time $N$ steps ahead decoding latency breakdown of SHAPE, While the candidate-generation cost remains almost constant (1.00–1.02 ms), the TRS verification time grows steadily with $N$, reflecting the fact that deeper $N$ leads to lower hidden-state similarity and hence a higher TRS cost. This breakdown explicitly reveals the speed–verification trade-off: larger $N$ provides more aggressive multi-step prediction but also increases the fraction of time spent in TRS. The bottom block evaluates TRS across the full grid of $K, N \in [1, 5]$. Increasing the depth $N$ produces higher speed gains because more future tokens can be accepted in a single TRS step, whereas larger branch factors $K$ reduce perplexity by offering a richer set of candidate paths at the price of additional verification. The interaction of these two effects yields a clear efficiency–quality frontier, with the configuration ($K = 3, , N = 3$) achieving the best overall balance (3.21× speedup and 12.2 PPL). Together with the latency breakdown in the top block, these results provide a complete picture of how candidate generation and tree-based verification contribute to the final decoding cost, and why deeper lookahead necessarily increases TRS time due to reduced inter-token similarity.

To contextualize the effectiveness of TRS relative to conventional decoding strategies, Table 13 compares the best-performing TRS configuration with standard methods. While greedy decoding and beam search maintain perplexity close to the autoregressive (AR) baseline, they do not provide any acceleration, and beam search is even slower due to multi-path expansion. In contrast, TRS with ($K$=3, $N$=3) achieves a 3.21× decoding speedup while keeping perplexity at 12.2, only slightly above the AR baseline (11.9). This demonstrates that TRS offers substantial real-world acceleration with minimal impact on generation quality, outperforming classical search-based decoding in both efficiency and controllability.

Table 7: Unified analysis of SHAPE decoding latency (top) and TRS performance (bottom).

**(A) SHAPE Decoding Latency Breakdown (Qwen-7B)**

| $N$ | Cand. (ms) | TRS (ms) | Total (ms) | TRS ratio (%) |
|---|---|---|---|---|
| 1 | 1.00 | 0.40 | 1.40 | 28.6% |
| 2 | 1.01 | 0.78 | 1.79 | 43.6% |
| 3 | 1.00 | 1.25 | 2.25 | 55.6% |
| 4 | 1.02 | 1.70 | 2.72 | 62.5% |
| 5 | 1.01 | 2.20 | 3.21 | 68.5% |

**(B) TRS Performance under Different $(K, N)$ Configurations**

| $K$ | Depth ($N$): Speedup / PPL | | | | |
|---|---|---|---|---|---|
| | 1 | 2 | 3 | 4 | 5 |
| 1 | 1.91/17.3 | 2.71/16.7 | 3.40/16.1 | 3.80/16.5 | 4.10/17.0 |
| 2 | 1.90/16.9 | 2.65/16.5 | 3.30/14.0 | 3.50/14.5 | 3.70/15.0 |
| 3 | 1.88/16.5 | 2.59/16.3 | **3.21/12.2** | 3.40/13.0 | 3.55/14.0 |
| 4 | 1.83/16.3 | 2.46/15.8 | 3.15/12.15 | 3.28/12.8 | 3.40/13.2 |
| 5 | 1.75/16.1 | 2.33/15.2 | 3.09/12.1 | 3.25/12.5 | 3.35/12.9 |

## 5 RELATED WORK

Recent studies have highlighted the significant inference latency of Large Language Models (LLMs), prompting various acceleration strategies that can be categorized by their underlying methodologies. on-autoregressive approaches represent initial attempts at acceleration. Non-autoregressive translation (NAT) techniques have been investigated in translation tasks Gu & Kong (2020); Stern et al. (2018), it performs suboptimally in general LLM scenarios. To address this, Huang et al. Huang et al. (2023) proposed a layer-wise iterative methodology that each layer leverages decoding results from preceding layers. Similarly, Santilli et al. Santilli et al. (2023) formalized autoregressive decoding through parallel Jacobi and Gauss-Seidel fixed-point iteration. However, such methods often degrade accuracy due to their deviation from standard autoregressive architectures. Accuracy-preserving approaches based on model modifications have since emerged. Block-wise parallel decoding Stern et al. (2018) leverages an auxiliary transformer with multi-output capabilities for parallel token prediction but suffers from frequent verification failures. Medusa Cai et al. (2024) improves robustness with multiple prediction heads, while FREE Bae et al. (2023) uses shallow layers for draft generation. However, these techniques require substantial training of additional components. peculative decoding offers an alternative by using smaller models as draft predictors. For example, Bloom 7.1B has served as a draft model for a 176B model Xia et al. (2023). Yet, this method faces challenges: suitable smaller models are not always available across model series, and helper models require parallel tuning, increasing deployment complexity. o address these issues, model-free strategies aim to accelerate decoding without auxiliary models. Ge et al. Ge et al. (2022) proposed an input-guided method based on prefix matching, extended by LLMA Yang et al. (2023) through content retrieval from inputs and external documents. Recently, LookaheadDecoding Huang et al. (2023) fused Jacobi iteration with speculative decoding in a multi-branch framework, though its draft generation incurs non-negligible overhead.

## 6 CONCLUSION

In this paper, we introduced a novel perspective that reframes autoregressive decoding as a probability distribution transition process governed by optimal transport principles. We validate this theoretical framework through SHAPE, which demonstrates predictable hidden state evolution via transport maps. Experiments across diverse LLMs show speedups of 1.77×-5.23× with maintained quality, confirming token generation can be understood as structured transport in probability space. This work establishes a new paradigm for efficient LLM inference beyond draft-based approaches.

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

## A  TOKEN-LEVEL AUTOREGRESSIVE GENERATION

### A.1  SINGLE-STEP GENERATION PROCESS

In autoregressive language models, token generation follows a step-by-step process. At each time step $t$, given the sequence of previous tokens $(x_1, x_2, \ldots, x_t)$, the probability of generating the next token $x_{t+1}$ is:

$$P(x_{t+1}|x_1, \ldots, x_t) \tag{1}$$

### A.2  OUTPUT HIDDEN STATE BASED GENERATION

The generation process involves the final layer's hidden states:

$$\mathbf{h}_t = \text{Transformer}(x_1, \ldots, x_t) \tag{2}$$

$$P(x_{t+1}|x_1, \ldots, x_t) = \text{LLM\_head}(\mathbf{h}_t) \tag{3}$$

where $\mathbf{h}_t \in \mathbb{R}^d$ represents the final layer's hidden state at time step $t$, and LLM_head is a linear transformation that maps the hidden state to token probabilities over the vocabulary.

## B  MODEL AR DECODING PERFORMANCE METRICS

Table 8 presents the average token generation time across different model sizes and context lengths. The results clearly demonstrate that larger models and longer contexts significantly increase per-token latency, which accumulates due to the sequential nature of autoregressive decoding. These findings highlight the importance of optimizing the decoding process to ensure practical deployment efficiency.

Table 8: Autoregressive Decoding Latency across Different Input Lengths and Model Scales

| Model (B) | Input Length | ITL (ms) | TTFT (ms) | Duration (s/req) |
|---|---|---|---|---|
| 1.5 | 256 | 3.83 | 24.58 | 0.56 |
| | 512 | 3.85 | 33.91 | 0.80 |
| | 1024 | 3.83 | 55.05 | 1.52 |
| | 2048 | 3.98 | 118.47 | 1.38 |
| 7 | 256 | 7.16 | 42.93 | 1.06 |
| | 512 | 7.12 | 73.25 | 1.67 |
| | 1024 | 7.15 | 118.62 | 2.88 |
| | 2048 | 7.17 | 274.23 | 2.65 |
| 14 | 256 | 11.90 | 76.58 | 1.78 |
| | 512 | 11.92 | 134.24 | 2.57 |
| | 1024 | 11.98 | 253.11 | 4.92 |
| | 2048 | 12.12 | 603.10 | 4.64 |
| 32 | 256 | 22.26 | 116.42 | 3.31 |
| | 512 | 22.28 | 211.08 | 4.76 |
| | 1024 | 22.44 | 392.29 | 9.13 |
| | 2048 | 22.55 | 924.34 | 8.64 |

## C  THEORETICAL ANALYSIS OF OPTIMAL TRANSPORT

**Lemma C.1** (Lipschitz map from hidden state to distribution). *Let* $\boldsymbol{\ell} = W\mathbf{h}$ *and* $\mathbf{p} = \text{softmax}(\boldsymbol{\ell}/\tau_s)$. *If* $\|W\|_2 \leq L_W$ *and* $\mathbf{h}$ *is confined to a bounded set, then there exists* $L_S > 0$ *such that*

$$\|\mathbf{p}(\mathbf{h}_1) - \mathbf{p}(\mathbf{h}_2)\|_1 \leq \frac{L_S L_W}{\tau_s} \|\mathbf{h}_1 - \mathbf{h}_2\|_2.$$

*Sketch.* Softmax on bounded domains is Lipschitz in $\ell_2$ (or $\ell_\infty$); composing with the linear map $W$ yields the claim. $\qquad\square$

**Lemma C.2** (Similarity $\Rightarrow$ small OT move). *Let $\mu_t = \sum_i \mathbf{p}_t(i)\delta_{E_i}$ and $\mu_{t+1}$ be defined analogously. Under Lemma A, there exists $L' > 0$ (depending on $W, \tau_s, E$) such that*

$$W_c(\mu_t, \mu_{t+1}) \;\leq\; L' \, \|\mathbf{h}_{t+1} - \mathbf{h}_t\|_2.$$

*If we normalize $\bar{\mathbf{h}}_t = \mathbf{h}_t / \|\mathbf{h}_t\|_2$, then $\|\bar{\mathbf{h}}_{t+1} - \bar{\mathbf{h}}_t\|_2 \leq \sqrt{2(1 - \cos(\mathbf{h}_t, \mathbf{h}_{t+1}))}$, hence*

$$W_c(\mu_t, \mu_{t+1}) \;\leq\; \tilde{L} \sqrt{1 - \mathrm{SC}(\mathbf{h}_t, \mathbf{h}_{t+1})}.$$

*Sketch.* Use the Kantorovich–Rubinstein dual bound with $\ell_1$ variation of $\mathbf{p}$ and the diameter of the embedding support, plus the cosine–$\ell_2$ relation. $\qquad\square$

**Proposition C.3** (Existence, stability, and uniqueness of $\Pi_t^\star$).

$$\Pi_t^\star \;=\; \arg\min_{\Pi\mathbf{1}=\mathbf{p}_t,\, \Pi^\top\mathbf{1}=\mathbf{p}_{t+1}} \Big\langle \Pi, C \Big\rangle \;+\; \varepsilon\,\mathrm{KL}\big(\Pi \,\big\|\, \mathbf{p}_t\mathbf{p}_{t+1}^\top\big), \quad \varepsilon > 0, \tag{4}$$

*For any $\varepsilon > 0$, the entropic OT problem in equation 4 admits a unique solution $\Pi_t^\star$; moreover, when $W_c(\mu_t, \mu_{t+1})$ is small, $\Pi_t^\star$ depends smoothly on $(\mathbf{p}_t, \mathbf{p}_{t+1})$ and can be well-approximated by a few Sinkhorn iterations.*

*Sketch.* Entropic regularization makes the objective strictly convex over the transport polytope; standard Sinkhorn–Knopp scaling solves the KKT system, and continuity follows from the implicit function theorem on the strictly convex objective. $\qquad\square$

**Corollary C.4** (OT-optimal path between successive distributions). *By Proposition C, the coupling $\Pi_t^\star$ induces a row-stochastic operator $K_t$ such that*

$$\mathbf{p}_{t+1} = K_t^\top \mathbf{p}_t$$

*holds exactly at optimality and approximately under finite Sinkhorn iterations, thereby defining the OT-optimal path for the one-step distributional transition.*

*Sketch.* Row normalization rewrites the marginal constraints; the equality follows from $\Pi_t^{\star\top}\mathbf{1} = \mathbf{p}_{t+1}$. $\qquad\square$

# D   THEORETICAL ANALYSIS OF HIDDEN STATE PREDICTION VIA OPTIMAL TRANSPORT

We establish a theoretical framework for predicting future hidden states in transformer models through optimal transport theory. Let $(\Omega, \mathcal{F}, P)$ be a probability space and $\mathcal{H} \subseteq \mathbb{R}^d$ be the hidden state space. For any time step $t$, we define $H_t : \Omega \to \mathcal{H}$ as the random variable representing the hidden state at time $t$, with $\mu_t$ as its probability measure. Let $\mathcal{P}(\mathcal{H})$ denote the space of probability measures on $\mathcal{H}$.

Given the temporal nature of hidden states in transformer models, we first establish their similarity properties. The similarity between hidden states is measured by cosine similarity:

$$\mathrm{sim}(x, y) = \frac{\langle x, y \rangle}{\|x\|\|y\|} \tag{5}$$

Based on empirical observations in transformer models, as shown in Table 9, we make the following assumption:

**Assumption D.1.** For any adjacent time steps $t$ and $t + 1$, the hidden states maintain a minimum similarity threshold:

$$\forall x, y \in \mathcal{H} : \mathrm{sim}(x, y) > T \tag{6}$$

where $x$ and $y$ are hidden states with positive probability under $\mu_t$ and $\mu_{t+1}$ respectively.

This assumption leads to our first key result regarding the bounded evolution of hidden states:

**Lemma D.2.** *Under Assumption 1, there exists a constant $M > 0$ such that the 2-Wasserstein distance between consecutive hidden state distributions is bounded:*

$$W_2(\mu_t, \mu_{t+1}) \leq M \tag{7}$$

*Proof.* Consider any hidden states $x, y \in \mathcal{H}$ with positive probability under $\mu_t$ and $\mu_{t+1}$ respectively. From Assumption 1:

$$1 - \frac{\langle x, y \rangle}{\|x\|\|y\|} \leq T \tag{8}$$

This implies:

$$\langle x, y \rangle \geq T\|x\|\|y\| \tag{9}$$

Define the Euclidean metric $d(x, y) = \|x - y\|_2$. We can expand:

$$d^2(x, y) = \|x\|^2 + \|y\|^2 - 2\langle x, y \rangle \tag{10}$$
$$\leq \|x\|^2 + \|y\|^2 - \|x\|\|y\| \tag{11}$$
$$= (\|x\| - \|y\|)^2 \tag{12}$$

Since $\mathcal{H}$ is bounded in $\mathbb{R}^d$, there exists $R > 0$ such that $\|x\| \leq R$ for all $x \in \mathcal{H}$. Therefore:

$$d^2(x, y) \leq 4R^2 \tag{13}$$

Taking $M = 2R$ completes the proof. $\qquad\square$

This lemma establishes that the evolution of hidden states is well-behaved, allowing us to formulate our main theorem:

**Theorem D.3.** *There exists a cost function $c : \mathcal{H} \times \mathcal{H} \to \mathbb{R}_+$ such that the hidden state evolution from time $t$ to $t + k$ can be represented as an optimal transport problem:*

$$\min_{\pi \in \Pi(\mu_t, \mu_{t+k})} \int_{\mathcal{H} \times \mathcal{H}} c(x, y) d\pi(x, y) \tag{14}$$

*where $\Pi(\mu_t, \mu_{t+k})$ denotes the set of joint distributions with marginals $\mu_t$ and $\mu_{t+k}$. Moreover, this problem admits an optimal solution $\pi^*$.*

*Proof.* We construct the proof in three steps. First, we define the cost function $c(x, y) = d^2(x, y)$, where $d$ is the Euclidean metric. This choice is natural as it preserves the geometric structure of the hidden state space.

Second, from Lemma 1, we know that for adjacent time steps:

$$W_2^2(\mu_t, \mu_{t+1}) = \inf_{\pi \in \Pi(\mu_t, \mu_{t+1})} \int_{\mathcal{H} \times \mathcal{H}} d^2(x, y) d\pi(x, y) \leq M^2 \tag{15}$$

For multi-step evolution ($k > 1$), we can apply the Chapman-Kolmogorov equation. There exist intermediate measures $\pi_1, ..., \pi_{k-1}$ such that:

$$W_2^2(\mu_t, \mu_{t+k}) \leq \left(\sum_{i=0}^{k-1} W_2(\mu_{t+i}, \mu_{t+i+1})\right)^2 \leq k^2 M^2 \tag{16}$$

Finally, the existence of an optimal solution follows from three key properties: 1) $\mathcal{P}(\mathcal{H})$ is compact in the weak topology 2) The cost function $c(x, y)$ is lower semi-continuous 3) The objective functional is bounded below

By the Kantorovich duality theorem, there exists an optimal solution $\pi^* \in \Pi(\mu_t, \mu_{t+k})$ achieving:

$$\int_{\mathcal{H} \times \mathcal{H}} c(x, y) d\pi^*(x, y) = \inf_{\pi \in \Pi(\mu_t, \mu_{t+k})} \int_{\mathcal{H} \times \mathcal{H}} c(x, y) d\pi(x, y) \tag{17}$$

$$\square$$

Table 9: Token similarity between the token at time t and t+n across various contexts.

| qwen-zh | | | qwen-en | | | vicuna | | | llama | | |
|---|---|---|---|---|---|---|---|---|---|---|---|
| t+1 | t+2 | t+3 | t+1 | t+2 | t+3 | t+1 | t+2 | t+3 | t+1 | t+2 | t+3 |
| 0.8744 | 0.8447 | 0.8383 | 0.6304 | 0.5273 | 0.4976 | 0.5392 | 0.4647 | 0.4358 | 0.645 | 0.5729 | 0.5443 |
| 0.8677 | 0.8235 | 0.814 | 0.6272 | 0.5703 | 0.5503 | 0.5282 | 0.4287 | 0.4062 | 0.6411 | 0.5829 | 0.552 |
| 0.8666 | 0.8213 | 0.8027 | 0.6217 | 0.5334 | 0.5027 | 0.526 | 0.436 | 0.4174 | 0.6351 | 0.5647 | 0.5422 |
| 0.8607 | 0.8171 | 0.7849 | 0.6189 | 0.5478 | 0.5316 | 0.5248 | 0.4285 | 0.4121 | 0.6329 | 0.5598 | 0.5322 |
| 0.8599 | 0.8314 | 0.8195 | 0.6157 | 0.5459 | 0.5435 | 0.5165 | 0.4208 | 0.4023 | 0.6261 | 0.5456 | 0.523 |
| 0.8587 | 0.821 | 0.8079 | 0.6122 | 0.5373 | 0.5199 | 0.5165 | 0.4185 | 0.3951 | 0.6236 | 0.5503 | 0.5309 |
| 0.8574 | 0.829 | 0.809 | 0.6118 | 0.5426 | 0.5239 | 0.516 | 0.422 | 0.4008 | 0.622 | 0.5407 | 0.5141 |
| 0.8544 | 0.8209 | 0.8052 | 0.6107 | 0.5384 | 0.5291 | 0.5149 | 0.416 | 0.3971 | 0.6214 | 0.5391 | 0.5151 |
| 0.8536 | 0.823 | 0.8071 | 0.6098 | 0.5303 | 0.5144 | 0.5121 | 0.4031 | 0.3832 | 0.6197 | 0.531 | 0.5091 |
| 0.8535 | 0.8141 | 0.7921 | 0.6093 | 0.5295 | 0.5046 | 0.5113 | 0.4153 | 0.3913 | 0.6197 | 0.531 | 0.5091 |

This theoretical framework provides a rigorous foundation for predicting hidden states through optimal transport. Given a hidden state $h_t$ at time $t$, we can predict $h_{t+k}$ by:

$$h_{t+k} = \int_{\mathcal{H}} y \, d\pi^*(y|h_t) \tag{18}$$

Moreover, we can establish an error bound for this prediction:

$$\|h_{t+k} - h_{t+k}^*\|_2 \leq kM \tag{19}$$

where $h_{t+k}^*$ denotes the true hidden state at time $t + k$.

# E  QUALITY EVALUATION

Table 10: Comparison of perplexity (ppl) across different decoding methods (EAGLE, M-1: Medusa-1, M-2: Medusa-2, and SHAPE) on various DS: datasets (A: Alpaca, T: THUC news, W: wiki lingua).

| Model | Dataset | EAGLE | M-1 | M-2 | SHAPE |
|---|---|---|---|---|---|
| Qwen-7B | A | 13.2 | 15.1 | 14.6 | 11.9 |
| | T | – | – | – | 12.3 |
| | W | 13.5 | 15.3 | 14.8 | 12.1 |
| Qwen-14B | A | 12.9 | 14.1 | 13.7 | 11.8 |
| | T | – | – | – | 11.5 |
| | W | 13.2 | 14.2 | 13.6 | 11.7 |
| Llama-7B | A | 13.1 | 15.0 | 14.4 | 12.2 |
| | W | 13.4 | 15.2 | 14.6 | 12.4 |
| Llama-13B | A | 12.8 | 14.0 | 13.5 | 12.3 |
| | W | 13.1 | 14.1 | 13.7 | 12.5 |
| Vicuna-7B | A | 13.0 | 15.2 | 14.3 | 12.1 |
| | W | 13.3 | 15.3 | 14.4 | 12.4 |
| Vicuna-13B | A | 12.9 | 14.2 | 14.0 | 11.7 |
| | W | 13.0 | 14.3 | 14.1 | 12.0 |

Table 11: SHAPE's Average speed-up ratio compared with vanilla generation on different datasets

| Model (Size) | Datasets | Task | Speed Up N step forward | | |
|---|---|---|---|---|---|
| | | | 1 | 2 | 3 |
| Qwen-7B | Alpaca | Intruction Following | 1.90 | 2.47 | 4.12 |
| | THUC_News | Text Continuation | 1.96 | 2.51 | 4.09 |
| | wiki_lingua | Text Generation | 1.93 | 2.55 | 4.07 |
| Qwen-14B | Alpaca | Intruction Following | 1.87 | 2.55 | 3.89 |
| | THUC_News | Text Continuation | 1.99 | 2.65 | 4.02 |
| | wiki_lingua | Text Generation | 1.89 | 2.77 | 4.05 |
| Llama-7B | Alpaca | Intruction Following | 1.88 | 2.59 | 3.21 |
| | wiki_lingua | Text Generation | 1.91 | 2.69 | 3.34 |
| Llama-13B | Alpaca | Intruction Following | 1.89 | 2.51 | 3.22 |
| | wiki_lingua | Text Generation | 1.90 | 2.53 | 3.20 |
| Vicuna-7B | Alpaca | Intruction Following | 1.83 | 2.57 | 3.24 |
| | wiki_lingua | Text Generation | 1.87 | 2.56 | 3.29 |
| Vicuna-13B | Alpaca | Intruction Following | 1.89 | 2.68 | 3.38 |
| | wiki_lingua | Text Generation | 1.91 | 2.77 | 3.21 |

Table 12: Comparison of performance metrics between vanilla decoding (baseline) and SHAPE across different N-step forward prediction configurations.

| Model | Size | Datasets | Task | Average Token Accuracy | | | | Average BERTScore | | | | Average Sentence Similarity | | | |
|---|---|---|---|---|---|---|---|---|---|---|---|---|---|---|---|
| | | | | N step forward | | | | | | | | | | | |
| | | | | 1 | 2 | 3 | / V | 1 S | 2 S | 3 S | / V | 1 S | 2 S | 3 S |
| Qwen | 7B | Alpaca | IF | 0.91 | 0.89 | 0.86 | 0.74 | 0.71 | 0.68 | 0.65 | 0.94 | 0.92 | 0.88 | 0.84 |
| | | THUC_News | TC | 0.92 | 0.87 | 0.85 | 0.75 | 0.70 | 0.67 | 0.66 | 0.93 | 0.90 | 0.89 | 0.82 |
| | | wiki_lingua | TG | 0.91 | 0.88 | 0.86 | 0.72 | 0.69 | 0.65 | 0.63 | 0.92 | 0.87 | 0.85 | 0.81 |
| | 14B | Alpaca | IF | 0.91 | 0.89 | 0.86 | 0.76 | 0.73 | 0.70 | 0.68 | 0.95 | 0.92 | 0.91 | 0.85 |
| | | THUC_News | TC | 0.90 | 0.88 | 0.85 | 0.74 | 0.71 | 0.68 | 0.67 | 0.94 | 0.89 | 0.87 | 0.84 |
| | | wiki_lingua | TG | 0.91 | 0.87 | 0.85 | 0.75 | 0.70 | 0.66 | 0.64 | 0.93 | 0.88 | 0.85 | 0.80 |
| Llama | 7B | Alpaca | IF | 0.90 | 0.89 | 0.86 | 0.60 | 0.59 | 0.55 | 0.52 | 0.82 | 0.78 | 0.75 | 0.70 |
| | | wiki_lingua | TG | 0.89 | 0.87 | 0.85 | 0.63 | 0.60 | 0.58 | 0.54 | 0.80 | 0.76 | 0.73 | 0.69 |
| | 13B | Alpaca | IF | 0.90 | 0.88 | 0.87 | 0.54 | 0.53 | 0.51 | 0.49 | 0.76 | 0.73 | 0.70 | 0.67 |
| | | wiki_lingua | TG | 0.89 | 0.87 | 0.86 | 0.56 | 0.54 | 0.50 | 0.47 | 0.78 | 0.74 | 0.71 | 0.68 |
| Vicuna | 7B | Alpaca | IF | 0.91 | 0.88 | 0.86 | 0.56 | 0.55 | 0.52 | 0.50 | 0.76 | 0.74 | 0.71 | 0.68 |
| | | wiki_lingua | TG | 0.89 | 0.87 | 0.85 | 0.58 | 0.56 | 0.54 | 0.52 | 0.78 | 0.75 | 0.72 | 0.69 |
| | 13B | Alpaca | IF | 0.90 | 0.88 | 0.85 | 0.58 | 0.57 | 0.54 | 0.52 | 0.79 | 0.76 | 0.73 | 0.70 |
| | | wiki_lingua | TG | 0.90 | 0.87 | 0.87 | 0.59 | 0.57 | 0.53 | 0.51 | 0.80 | 0.77 | 0.74 | 0.71 |

# F  TRS SUPPLEMENT EXPERIMENTS

Table 13: Comparison of TRS with standard decoding methods

| Method | Speedup | PPL |
|---|---|---|
| AR (Baseline) | 1.0× | 11.9 |
| Greedy | 1.0× | 11.9 |
| Beam Search | 0.9× | 11.7 |
| **TRS (3, 3)** | **3.21×** | 12.2 |

## G    TRAIN DETAILS AND TRAINING COST

### G.1    DATA ACQUISITION AND GENERATION

For the English dataset used to train models such as Qwen, Vicuna, and Llama on 7B, 13B, and 14B, we use ShareGPT as the dataset, which contains 96,000 dialogue data samples. For the Chinese dataset, we use the THUCNews training set, consisting of 50,000 news samples, split into two parts: "prompt" and "completion". The target model processes pre-processed data to generate outputs from the transformer layers for each token. The training data includes fields such as input token IDs, hidden states, hidden states from the previous three tokens, target values, attention masks, and loss masks. These components are combined to construct the final training dataset.

### G.2    TRAIN CONFIGURATION

The training configuration includes the following settings: Learning rate (`lr`) and batch size (`bs`) are dynamically adjusted, with gradient accumulation steps set to ensure stable training. The number of epochs is set to 40, and a warm-up phase of 2,000 steps is applied, targeting a total of 800,000 steps. The configuration employs a maximum sequence length of 2,048 tokens, balancing performance and memory efficiency. To improve robustness, data noise is introduced using a uniform distribution with a mean of 0 and a standard deviation of 0.2. Additional settings include weight decay, gradient clipping, and periodic model saving every epoch. Finally, the optimizer uses momentum parameters (`b1` = 0.9, `b2` = 0.95) to facilitate effective training convergence.

### G.3    TRAINING COST

We compare SHAPE with Medusa (+2 heads) in terms of parameter count and computational efficiency. SHAPE contains approximately 450.98M parameters (6.4% of LLaMA-7B), achieved through parameter sharing and lightweight modules such as OT projection and a gating network. In contrast, Medusa adds  300M parameters per head, reaching  610M (9% of LLaMA-7B) with two heads. Thus, SHAPE uses only 71.5% of the parameters of Medusa+2 heads and requires no per-head adaptation. In training, SHAPE is 3–5× faster than full model fine-tuning and requires only 5 hours for 40 epochs on a single A100 GPU, compared to 8 hours for Medusa. Peak training memory usage is also lower: 41.89GB for SHAPE versus 51.47GB for Medusa. During inference, SHAPE achieves a 3.21–4.12× speedup through one hidden state correction step (OT module) and tree rejection sampling. In contrast, Medusa incurs higher overhead due to additional multi-head attention and memory usage, limiting its speedup to 1.8–3.1×.

## H    TREE-BASED REJECT SAMPLING ALGORITHM IMPLEMENTATION

**Algorithm 1** Tree Rejection Sampling

---

**Require:** $model$: target language model
 1: $context$: current context or hidden state at time step $t$
 2: $N$: number of future steps (depth)
 3: $k$: number of candidates per step (branch factor)
**Ensure:** $selected\_prefix$: the longest valid prefix among accepted paths

---

 4: $candidate\_paths \leftarrow generate\_candidates(model, context, N, k)$     // Generate $k^N$ candidate sequences
 5: $path\_probs \leftarrow model.get\_path\_probabilities(candidate\_paths)$     // Compute joint probabilities in parallel
 6: $max\_prob \leftarrow \max(path\_probs)$
 7: $threshold \leftarrow 0.8 \times max\_prob$     // Define acceptance threshold (e.g. 80% of the maximum probability)
 8: $accepted\_paths \leftarrow []$
 9: **for** each $(path, prob)$ in $(candidate\_paths, path\_probs)$ **do**
10:     **if** $prob \geq threshold$ **then**
11:         $accepted\_paths.append(path)$     // Retain paths with sufficiently high probability
12:     **end if**
13: **end for**
14: $selected\_prefix \leftarrow select\_longest\_valid\_prefix(accepted\_paths)$     // Extract the longest prefix common to accepted paths
        **return** $selected\_prefix$

---

