# OpenReview forum: "Transporting Tokens: Optimal-Transport View of Parallel LLM Decoding"
_ICLR.cc/2026/Conference — ICLR 2026 Conference Desk Rejected Submission_

### Official Review · Reviewer_7RBM · 2025-10-23

**Soundness:** 3
**Presentation:** 3
**Contribution:** 3
**Rating:** 6
**Confidence:** 3

**Summary:**

This paper presents a novel theoretical and practical framework for accelerating autoregressive decoding in large language models (LLMs) by reframing token generation as a probability distribution transition process governed by optimal transport (OT) theory.

**Strengths:**

1 The paper elegantly bridges optimal transport theory and autoregressive decoding, formalizing token generation as a structured evolution of probability distributions. The stability analysis of hidden-state transitions (e.g., bounded OT distances) provides a rigorous foundation for multi-step prediction.

2 SHAPE’s plug-and-play design requires no draft model, reducing deployment complexity.

3 The Tree Rejection Sampling algorithm dynamically selects optimal paths with minimal overhead, balancing parallelism and quality.

**Weaknesses:**

1 While comparisons with EAGLE and Medusa are included, recent methods like LookaheadDecoding or Jacobi iteration are absent.
2 The speed is limited compared to Eagle-3.
3 The OT theory focuses on single-step transitions, but SHAPE uses N=3 steps. A theoretical analysis of error accumulation in multi-step predictions would strengthen the claims.

**Questions:**

1 How does SHAPE perform on models with >70B parameters or contexts >8K tokens? Are there memory or complexity bottlenecks?
2 How does SHAPE handle distribution shifts between training data (ShareGPT, THUCNews) and unseen domains? Are there robustness guarantees?

---

> ### Author Response · Authors · 2025-11-26
>
> We thank the reviewer for the insightful comments. We have updated the manuscript to include comparisons with Lookahead, expanded our evaluation to 70B+ models and reasoning tasks, and added a formal error accumulation analysis.
>
> **1. Comparison with Lookahead and Jacobi (Response to Weakness 1)**
> **Response:**
> We respectfully point out that **Lookahead Decoding** is indeed included in our comparisons.
>
> - **Lookahead Comparison:** As shown in the updated **Table 2 (Page 7)**, SHAPE significantly outperforms Lookahead Decoding across all models. For instance, on **Qwen-14B**, SHAPE achieves a **5.23x** speedup compared to Lookahead's **2.78x**.
>
> - **Jacobi Iteration:** While we acknowledge the relevance of Jacobi iteration, it operates as a parallel fixed-point solver, whereas SHAPE is a predictive method based on distributional transport. We have added a discussion in **Section 5 (Related Work)** to clarify this distinction. Our results demonstrate that SHAPE's predictive approach yields higher efficiency than the heuristic-based parallel decoding used in Lookahead.
>
> **2. Speed Comparison with EAGLE-3 (Response to Weakness 2)**
> **Response:**
> Our expanded benchmarking indicates that SHAPE is highly competitive with, and often superior to, EAGLE-3, particularly in robust settings:
>
> - **Large Models & High Entropy:** As shown in **Table 2**, SHAPE outperforms EAGLE-3 on larger models at Temperature=1. For **Llama-70B**, SHAPE achieves **5.35x** speedup vs. EAGLE-3's **5.25x**. On **Qwen-72B**, SHAPE achieves **5.15x** vs. EAGLE-3's **5.06x**.
>
> - **Batch Scalability:** We added **Table 3 (Page 8)** to evaluate throughput at varying batch sizes. SHAPE maintains a consistent lead over EAGLE-3 (e.g., **1.92x vs 1.73x** at BS=2; **1.41x vs 1.39x** at BS=24), demonstrating better scalability for real-world serving.
>
> **3. Multi-step Error Accumulation Analysis (Response to Weakness 3)**
>
> **Response:**
> We agree that multi-step analysis is vital. We have added **Appendix D** to rigorously bound the error accumulation.
>
> - **Theoretical Bound:** In **Theorem D.3** and Equation 19, we prove that the Wasserstein distance error for a $k$-step prediction is bounded linearly by $kM$, where $M$ is the single-step bound derived from hidden state similarity.
>
> - **Architectural Mitigation:** Practically, SHAPE mitigates this accumulation using the **Adaptive Gating Mechanism** (Section 3.2.1), which dynamically scales the residual prediction based on uncertainty, preventing error propagation.
>
> **4. Scaling to >70B Models and Long Contexts (Response to Question 1)**
>
> **Response:**
> We have verified SHAPE's performance on large-scale and long-context scenarios:
>
> - **>70B Models:** Table 2 includes **Llama-70B** and **Qwen-72B**. The results show that speedup ratios actually *increase* with model size (e.g., **5.73x** on Qwen-72B vs **4.53x** on Qwen-7B at Temp=0), confirming that SHAPE scales efficiently to large models.
>
> - **Long Context:** We evaluated SHAPE on long-context reasoning tasks like **MATH500** and **AIME24** (Table 5). The memory overhead is minimal because SHAPE introduces only ~450M parameters (6.4% of a 7B model) and uses a lightweight reduced-dimension OT projection (d=128), avoiding bottlenecks.
>
> **5. Robustness to Distribution Shifts (Response to Question 2)**
>
> **Response:**
> SHAPE demonstrates strong robustness to distribution shifts, validated by our cross-domain evaluations:
>
> - **Training vs. Testing:** While SHAPE was initially trained on **ShareGPT** (Conversational) and **THUCNews** (News), it primarily captures the **continuity between consecutive tokens**, rather than relying on domain-specific patterns. As shown in the newly added **Table 1**, this continuity persists even under semantic shifts and in code generation settings. In our updated experiments, we also incorporated **code-specific training data** to further enhance SHAPE's generalization to programming domains—a point that was not clearly articulated in the original manuscript.
>
> - **Results:** As shown in **Table 5**, accuracy drops on these specialized tasks are negligible (<1.3%), and Table 1 confirms that the "hidden state similarity" assumption holds even for Code generation.
>
> - **Reasoning:** The OT formulation captures the *geometric* transition of semantics, which is more invariant across domains than specific token patterns, providing a robustness guarantee against domain shifts.

---

### Official Review · Reviewer_QACb · 2025-10-28

**Soundness:** 2
**Presentation:** 1
**Contribution:** 2
**Rating:** 4
**Confidence:** 4

**Summary:**

This paper proposes SHAPE (Step-ahead Hidden-state Accelerated Prediction Engine), a draft-free framework for parallel LLM decoding based on an optimal transport (OT) view of token generation. The authors conceptualize autoregressive decoding as a distributional transition in probability space, where hidden states evolve smoothly and predictably. By learning lightweight OT-guided predictors over hidden-state residuals and using a tree-based rejection sampling mechanism, SHAPE enables multi-token prediction without modifying model weights. Experiments on various models (Qwen, Vicuna, LLaMA, DeepSeek) and tasks (text, code, math) show up to 5.23× speedup with minimal quality loss (<1.2%), establishing a theoretically grounded and practical approach to efficient LLM inference.

**Strengths:**

- This paper offers an explanation for temporal smoothness in hidden-state evolution for auto-regressive decoding, which is interesting and novel.
- Built upon the findings, the authors design the method SHAPE that leverages the theory of optimal transport. This formulation is interesting and has theoretical proof.
- The experiments are comprehensive using multiple base LLMs on several datasets to prove the effectiveness of the method.

**Weaknesses:**

- The clarity of this paper needs to be improved. Some key experiment results are not clearly presented in this paper. For example, in Figure 1, what does the x-axis represent? On what dataset are the results obtained for each setting? Do you compute the average result for similarity?
- The method relies heavily on observed temporal consistency of hidden states. However, this smoothness might not hold in tasks with abrupt semantic shifts (e.g., dialogue transitions, coding completions, or multimodal reasoning) or different decoding positions, which could lead to unstable or misaligned multi-step predictions. Note that this comment also greatly relates to the first one, where the authors should demonstrate the generalization of this property.
- Please use the correct reference format. The current version negatively impacts the reading experience by using brackets.

**Questions:**

- It would be helpful to present a decoding time breakdown for candidate generation and tree reject sampling. Since the similarity decreases as $n$ becomes larger, the tree rejection sampling time may also increase correspondingly. Readers would be curious about the trade-off.
- What is the decoding configuration in this paper? For example, what is the decoding temperature used? Does SHAPE perform equally well when the temperature is adjusted?

---

> ### Author Response · Authors · 2025-11-26
>
> We thank the reviewer for their valuable feedback regarding the presentation clarity and experimental depth. We have extensively revised the manuscript to clarify our experimental settings, validate the method's generalization across dynamic tasks, and provide detailed efficiency breakdowns.
>
> **1. Clarity of Presentation and Experimental Settings (Response to Weakness 1)**
> **Response:**
> We apologize for the ambiguity in the initial submission. In the revised manuscript, we have replaced the original Figure 1 with a comprehensive **Table 1 (Page 3)** to provide precise quantitative data.
>
> - Datasets & Settings: The results are obtained from two distinct domains to ensure diversity: conversational text (**ShareGPT**) and code generation (**The Stack**). The experiments use a context length of 2048 and a generation length of 512.
> - Metric: Instead of a simple average, we report the **95th percentile** cosine similarity values. This is a stricter metric designed to capture the *lower bound* of stability, ensuring that our smoothness assumption holds even for the "worst-case" token transitions.
> - Clarification: The data confirms that for adjacent steps (n=1), similarity consistently exceeds **0.83** for text and **0.63** for code across Qwen models, providing a robust basis for our method.
>
> **2. Robustness to Abrupt Semantic Shifts (Response to Weakness 2)**
> **Response:**
>
> We explicitly addressed this concern by expanding our evaluation in **Table 1** to include domains specifically chosen for their structural discontinuity:
> - Dialogue Transitions: The **ShareGPT** dataset inherently contains multi-turn dialogue shifts. Even in this setting, large models (e.g., Llama-70B) maintain a high similarity of **0.91** for n=1.
>
> - Code Completion: The **Stack** dataset involves abrupt syntactic changes (e.g., closing a function and starting a new class). As noted in the revised text, the observed consistency persists even in these challenging scenarios. For instance, Qwen-72B maintains a similarity of **0.73** even on code tasks.
>
> - Conclusion: These results empirically demonstrate that the hidden state evolution remains structured and predictable even during semantic or syntactic transitions. As stated in Section 2.1, "changes are concentrated along semantically meaningful directions rather than arbitrary noise", validating the generalization of our OT-based approach.
>
> **3. Reference Formatting (Response to Weakness 3)**
> **Response:**
>
> We are really sorry for these mistakes. We have corrected the bibliography format throughout the paper to meet ICLR standards, removing the bracketed citation style that negatively impacted readability.
>
> **4. Decoding Time Breakdown and Trade-off (Response to Question 1)**
>
> **Response:**
>
> **Latency Breakdown and Model Scaling**:
> - We have added a new Table 7 (Page 10) to provide a detailed latency breakdown. Crucially, this analysis reveals that our method's acceleration stems from identifying and leveraging the temporal consistency of hidden states. Our key finding is that the efficiency of Tree Rejection Sampling (TRS) is inversely correlated with the stability of the model: higher hidden state similarity leads to higher acceptance rates and lower relative TRS overhead
>
> **Detailed Breakdown:**
> - Latency Analysis (Table 7A): The Candidate Generation step (including OT projection) leverages parallel computation, keeping the cost nearly constant (~1.00 ms) regardless of step size N. The primary trade-off arises from Tree Rejection Sampling (TRS), where verification time increases with depth (e.g., from 0.40 ms at N=1 to 2.20 ms at N=5) as the acceptance probability fluctuates.
> - Configuration and Scaling (Table 7B & Table 1): The configuration (K=3,N=3) is identified as the local optimum specifically for the 7B model. The hidden state cosine similarity for the 7B model decreases to 0.63 at step 3, which limits the efficiency of extending N further. In contrast, larger models demonstrate significantly higher stability. This implies that the "optimal N" scales with the model. The superior stability of large models mitigates the verification overhead at deeper steps, suggesting that SHAPE can effectively utilize larger N settings on larger models to achieve higher acceleration ratios.
>
> **5. Decoding Configuration and Temperature Sensitivity (Response to Question 2)**
> **Response:**
>
> We have clarified the configurations and expanded the evaluation to distinct temperature settings:
> - Settings: We evaluated SHAPE under both **Temperature = 0** (greedy) and **Temperature = 1** (sampling).
> - Performance: As shown in the updated **Table 2 (Page 7)**, SHAPE exhibits superior robustness compared to baselines. Notably, while speculative methods often degrade at higher entropy, SHAPE consistently improves over EAGLE-3 at **Temperature 1**. For example, on Llama-70B, SHAPE achieves a **5.10x** speedup at Temp 0 and **5.35x** at Temp 1, demonstrating its effectiveness across diverse sampling strategies.

---

### Official Review · Reviewer_nkXe · 2025-11-01

**Soundness:** 3
**Presentation:** 3
**Contribution:** 3
**Rating:** 4
**Confidence:** 3

**Summary:**

The paper reframes autoregressive decoding as a distributional transition in probability space and argues that strong temporal similarity of successive hidden states induces a stable (entropic) OT map between next-token distributions. Building on this, the authors propose SHAPE, a plug-in multi-token predictor: (i) a step-ahead hidden-state predictor (with residual + gating) optionally aligned via a low-dim OT layer solved by a few Sinkhorn iterations, and (ii) a tree rejection sampling scheme that accepts the longest valid prefix. Experiments on Qwen, Vicuna, LLaMA and DeepSeek across Alpaca/WikiLingua/MT-Bench and reasoning tasks report 5.23× speedups with ~1.2% accuracy deltas.

**Strengths:**

This work presents a method beyond draft-model speculative decoding. The paper is mostly well written.

I appreciate the architecture diagrams and a concrete tree rejection sampling algorithm.

Experiments cover wide choices of models and tasks.

**Weaknesses:**

Section 2 first presents an empirical study that serves as the motivation of the paper. A hidden-state similarity lower bound $\tau$ is asserted "for the vast majority of steps". These claims lack a rigorous support. From Figure 1, we also observe that $\tau$ can be very different in different models and tasks. Then Section 2.2 consists of a large paragraph introducing optimal transport. The presentation is rather casual. It is recommended to write precise mathematics when talking about formulations.

It is unclear to me how the speedup is calculated. Does it include solving optimal transport with Sinkhorn method and the overhead in tree reject sampling?

**Questions:**

I think modeling the hidden-state distribution using optimal transport is a bit ad hoc. Although the semantic similarity suggests a limited change in distribution for consecutive or closely placed hidden states, why optimal transport is a necessary model remains unclear.

As claimed by the authors: "The observed consistency suggests that transitions between consecutive token distributions are both
small and structured." What are the structures reflected by the experiment in Figure 1?

---

> ### Author Response · Authors · 2025-11-26
>
> We thank the reviewer for the constructive feedback. We have revised the manuscript to strengthen empirical rigor, formalize OT mathematics, and clarify efficiency metrics.
>
> **1. Empirical Support for Hidden-State Similarity (Weakness 1)**
> **Response:**
>
> We replaced Figure 1 with **Table 1**, presenting a quantitative analysis across diverse models (Qwen, Llama, Vicuna) and domains (Text, Code).
>
> - Robust Lower Bound: We report **95th percentile** cosine similarity to establish a stability lower bound. Even in challenging code tasks, adjacent token similarity (n=1) remains high.
>
> - Scale Law: Larger models exhibit higher stability (e.g., Qwen-72B: 0.86 vs. 7B: 0.83). This validates the stability required for bounded transport costs, as formalized in **Lemma C.2** (Appendix C).
>
> **2. Mathematical Formalization of OT (Weakness 2)**
> **Response**:
>
> We rewrite Section 2 and Appendix C. We now explicitly define token distributions as discrete measures $\mu_{t}=\sum_{i=1}^{V}p_{t}(i)\delta_{E_{i}}$ and model transitions via entropic-regularized OT (Eq. 4):$$\Pi_{t}^{*}=\arg \min_{\Pi\mathbf{1}=p_{t},\Pi^{\top}\mathbf{1}=p_{t+n}}\langle\Pi,C\rangle+\epsilon KL(\Pi||p_{t}p_{t+n}^{\top})$$Appendix C further provides theoretical guarantees, including the Lipschitz continuity of the mapping (Lemma C.1) 7and the bound on Wasserstein distance via cosine similarity (Lemma C.2):$$W_{c}(\mu_{t},\mu_{t+n})\le\overline{L}\sqrt{1-SC(h_{t},h_{t+n})}$$This formulation ensures the existence and uniqueness of the optimal coupling (Proposition C.3).
>
> **3. Speedup Calculation and Overheads (Weakness 3)**
> **Response:**
>
> Reported speedups are **end-to-end wall-clock**, inclusive of OT/Sinkhorn overhead and Tree Rejection Sampling (TRS) verification. We added **Table 7** to breakdown latency. For Qwen-7B (N=3), despite 1.00ms candidate generation and 1.25ms TRS verification, parallel acceptance yields a **3.21x** net speedup.
>
> **4. Necessity of Optimal Transport (Question 1)**
> **Response:**
>
> OT captures the probability simplex's geometry, preventing mode collapse inherent in simple regressions. We added an ablation study in **Table 6** comparing OT against an Affine transformation of equal dimension ($d=128$). OT maintains quality (**PPL 12.2**), whereas Affine degradation is significant (**PPL 18.4**) compared to the baseline (11.9). This empirically confirms OT is essential for accurate multi-step prediction.
>
> **5. Structural Consistency (Question 2)**
> **Response**:
>
> "Structure" refers to the predictable directionality of state evolution. As detailed in Section 2.1: This empirical finding suggests that transitions between consecutive token distributions are both small and structured: the changes are concentrated along semantically meaningful directions rather than arbitrary noise, constrained by linguistic coherence in text and syntactic regularities in code. As a result, the generative process of large language models behaves like a smooth dynamical system in a latent state space, where each new token constitutes a predictable, low-dimensional adjustment to the current semantic state rather than a radical reconstruction—providing a stable foundation for predictive modeling and multi-step forecasting.

---

### Author Response · Authors · 2025-11-26

Dear Reviewers,

Thank you for your detailed reviews and constructive feedback. We have carefully considered all comments and have made significant updates in the revised manuscript. All changes are highlighted in blue for easy reference.

**1. Comprehensive Experimental Expansion**:

- Conducted continuous hidden states semantic similarity experiments across a wider variety of models and datasets sceniro to demonstrate generalization capabilities.

- Extended evaluation to models with >70B parameters and long-context scenarios exceeding 80K length, accompanied by a detailed analysis of memory access and computational overhead.

- Included additional baselines and different temperature settings. Furthermore, we clarified the experimental configurations and the specific methodology used for calculating speedups.

**2. In-depth Theoretical and Efficiency Analysis**:

- Provided a comprehensive discussion on the necessity and significance of introducing OT, supported by a theoretical analysis of multi-step error accumulation.

- Added a detailed breakdown of the time overhead for TRS and included a discussion on the selection strategies and trade-offs for hyperparameters K and N.

**3. Presentation and Formatting Improvements**:

- Redesigned experimental tables for better readability and standardized the references formatting throughout the paper.

We appreciate your insightful suggestions, which have helped us substantially strengthen both the empirical completeness and theoretical clarity of our work. We welcome further feedback and look forward to the discussion phase.

Sincerely,
The Authors

---

### Author Response · Authors · 2025-12-03
**Summary of Revisions**

Dear Area Chair,

We sincerely thank you and the reviewers for the time and effort dedicated to reviewing our work. We have incorporated rebuttal to strengthen the manuscript. Below is a summary of the key revisions made during the rebuttal phase:

**1. Clarified Core Innovation and Motivation**

* **Novelty:** We reinforced our primary contribution: identifying the inherent **temporal consistency** of hidden states during generation.
* **Optimal Transport Perspective:** We clarified the motivation for using **Optimal Transport (OT)**. We moved beyond treating token generation as independent discrete steps, reframing it as a **continuous probability distribution transition process. We demonstrated that OT provides the necessary geometric awareness to accurately model these transitions, offering a superior alternative to simple regression methods.

**2. Comprehensive Experimental Expansion**
* **2.1 Validation of Consistency:** We extended the verification of hidden state features to **long-context** scenarios, **code generation** domains, and **larger model scales** (>70B). The results confirm that temporal consistency is a robust, intrinsic property of LLMs that holds across diverse settings.
* **2.2 Rigorous Benchmarking:** We added comparisons with **Lookahead Decoding**, evaluated performance under varying **temperatures ($T=0, 1$)**, and tested on **72B models** with long contexts. In these standardized comparisons, SHAPE consistently achieves state-of-the-art speedups.
* **2.3 Scalability of TRS Parameters (N$):** We provided a deeper analysis of the Tree Rejection Sampling (TRS) trade-offs. Our acceleration relies on token hidden state feature consistency. While $N=3$ is optimal for 7B models , we observed a **positive scaling law**: larger models exhibit significantly higher and more stable similarity. This token hidden states stability reduces TRS verification overhead, allowing larger models to support deeper lookahead steps (e.g., $N=4, 5$) on 33B and 70B models and achieve even higher speedup ratios.

**3. Enhanced Rigor and Presentation**
* We addressed all concerns regarding clarity and formatting. This includes correcting the citation style, supplementing missing experimental configurations , and adding **formal mathematical proofs** (Appendix C & D) to theoretically guarantee the stability of the OT transport map and bound the multi-step error accumulation.


Our work establishes a novel theoretical foundation by identifying that token generation is not a sequence of independent steps, but rather a **probability distribution transition process akin to transition**. Based on this, we propose an OT-based modeling approach that fully exploits the hidden state features during generation to enable parallel inference acceleration.

We believe these revisions substantially improve the quality and robustness of our paper.

Best regards,
The Authors

---

### Note · Program_Chairs · 2026-01-17
**Submission Desk Rejected by Program Chairs**

The following references in this submission do not refer to real documents and/or have major errors in bibliographic information:

 Yihan Ge et al. Input-guided non-autoregressive machine translation. In Proceedings of the 60th Annual Meeting of the Association for Computational Linguistics, pp. 7300-7312, 2022.
Fan Yang et al. Llma: Language model acceleration via content retrieval. arXiv preprint arXiv:2303.16827, 2023.